# Multi-omic analysis reveals lipid dysregulation associated with mitochondrial dysfunction in parkinson's disease brain

Jenny Hällqvist [1], Christina E. Toomey [2,3,4], Rui Pinto [5,6,7], Tomas Baldwin[1], Ivan Doykov[1], Anna Wernick[4], Mesfer Al Shahrani[8,9], James R. Evans [3,4], Joanne Lachica[3], Simon Pope [8], Simon Heales[8], Simon Eaton [10], Kevin Mills [1], Sonia Gandhi [3,4] & Wendy E. Heywood [1] ✉

Parkinson's disease (PD) is an increasingly prevalent neurodegenerative disorder, largely sporadic in origin, with limited understanding of age- and region-specific lipid alterations in the human brain. Dysregulation of glycosphingolipid catabolism has been implicated in PD, yet comprehensive spatiotemporal profiling remains sparse. Here, we performed targeted lipidomics across eight anatomically distinct brain regions in post-mortem controls, mid-stage, and late-stage PD cases using high-precision tissue dissection. Each region displayed distinct lipid signatures, with several age-associated alterations—most notably in hexosylceramides, including glucosylceramide. In PD, glycosphingolipids were reduced in subcortical regions but elevated in cortical regions, particularly gangliosides, HexCer, and Hex2Cer, accompanied by increased sphingolipids and decreased phospholipids. The most pronounced mid-stage changes occurred in the putamen, where very long chain ceramide species and plasmalogen PE decreased, then normalising in late-stage disease. Lyso-phosphatidylcholine increased progressively throughout PD progression. Integrating proteomic data, we observed sphingomyelin levels associated with PD-related proteins, while dysregulated mitochondrial function correlated with antioxidant plasmalogens, long-chain ceramides, lyso-phosphatidylcholine, and HexCer in the putamen. These findings highlight region- and stage-specific lipid alterations in PD and their potential convergence with mitochondrial dysfunction.

Parkinson's Disease (PD) is becoming increasingly prevalent in our ageing population and is the most common neurodegenerative movement disorder, affecting 1–2% of people aged over 65, with an estimated > 6.1 million people affected worldwide[1].There has been a wealth of research into genetic and proteomic changes affected by PD[2–5], however it is becoming more and more apparent that dysregulated lipid pathways also play a role in the aetiology of the disease[6–9]. Importantly, it has been observed that α-synuclein aggregation is affected by lipids[9] and that α-synuclein inclusions have a high lipid content[10]. However, considering that the brain consists of 60 – 70% lipids by dry weight, there is little recent literature[11,12] to describe the normal human brain lipid profile, how it varies across different brain regions, age and disease state.

The relevance of lipid changes is more difficult to delineate than genes or proteins, and their functions are not yet completely understood. Lipids are highly heterogenous and have many molecular

species determined by carbon chain length, functional groups, and the presence of unsaturated sites. The relevance and function of the many species is yet to be understood, but in a broad sense, sphingolipids are highly enriched in the nervous system where they act as constituents of plasma membranes[13] and play a role in membrane fluidity, recognition, signalling, and many cellular functions[14]. The major sphingolipid, sphingomyelin, has been implicated in PD[15] particularly due to its role in α-synuclein exosome composition where it may be involved in mitigating α-synuclein spread[16]. The glycosphingolipid degradation pathway is a key pathway implicated in genetic PD with mutations in the *GBA1* gene being the most common genetic cause for PD. A comprehensive figure of the glycosphingolipid degradation pathway is given in Supplementary Results Figure S1. Phospholipids belong to another class of lipids which are enriched in the brain and integral to the formation of plasma membranes. Dysregulation of phospholipids has been implicated in Alzheimer's disease[17] as well as following brain injury[18], but they are also known to interact with α-synuclein[9].

To expand on the knowledge of lipid levels in the human brain, we have profiled multiple brain regions in sporadic mid-stage PD (Braak stage 3-4), and late-stage PD (Braak stage 5-6), exploring how the lipid profile changes across eight brain regions in relation to age and disease, and correlated these findings with proteomics data in a multi-omic approach. We applied targeted lipid panels consisting of the main non-cholesterol lipids that are abundant in the brain, including sphingolipids, glycosphingolipids, and phospholipids as well as their deacylated lyso-forms. We also included phosphatidylethanolamine plasmalogens, an ether glycerophospholipid derived from the peroxisome which is abundant in brain tissue and has been found to be reduced in lipid rafts in the frontal cortex of PD affected brains[19]. Table 1 lists the lipids and species included in this analysis.

## Results

### Lipid abundances differ between anatomical regions and are modulated by age in healthy brains

We first examined the group of control samples ($n = 107$, 38% F) to establish baseline levels of the analysed lipids in healthy, aged brains, not affected by PD. The average distribution of each lipid class showed that there were spatial differences in which respective lipids levels were stronger. Examining the lipid classes in detail, we found that ceramide was the lipid class changing the least between regions, with a variation of 15%, while lyso-phosphatidylethanolamine (lyso-PE) and n-hexosylceramide (the sum of glucosylceramide and galactosylceramide) varied the most, presenting coefficients of variation of 82% and 52%, respectively (Fig. 1A). Taking an unbiased approach in regards to region proximity or function, we applied hierarchical clustering to evaluate the correlation between lipid-levels across the brain to determine whether the regions shared similar abundance profiles (Fig. 1B). Three regional clusters formed, consisting of (i) frontal cortex (FrC), caudate (CA) and parahippocampus (PHp), (ii) cerebellum (CBM), cingulate cortex (CiC) and putamen (PU), and (iii) parietal (PtC) and temporal cortices (TCtx). The primary driving parameter for this clustering was found to be a general elevated abundance of both phospho- and sphingolipids in the FrC-CA-PHp cluster, while the CBM-CiC-PU cluster demonstrated elevated levels of ganglioside, especially prominent in CiC and PU. Lastly, the PtC-TCtx cluster was distinguished by lower levels of most lipids but lyso-phospholipids, and ceramide. The clustering also showed that sphingomyelin, n-hexosylceramide (where n represents the number of hexosyl groups) and *N*-acetylneuraminic acid presented similar changes, as did the diacylated phospholipids. We further projected the lipids in a Uniform Manifold Approximation and Projection (UMAP, Fig. 1C)[20] to obtain an overview of how the different lipid species and lipid classes related to

## Table 1 | Compound classes measured and corresponding internal standards used

| Analyte compounds | Abbreviation | Fatty acid chains | Internal standard |
|---|---|---|---|
| Ceramide | Cer d18:1 | 16-24 (+ OH 16-24) | d$_3$-ceramide d18:1/18:0 |
| Ganglioside GM1 | GM1 d18:1 | 16-24 | d$_3$-GM1 d18:1/16:0 |
| Ganglioside GM2 | GM2 d18:1 | 16-20 | d$_3$-GM2 d18:1/16:0 |
| Ganglioside GM3 | GM3 d18:1 | 18 | d$_3$-GM3 d18:1/16:0 |
| *N*-acetylneuraminic acid | NANA | n/a | d$_3$-GM1 d18:1/16:0 |
| Hexosylceramide (glucosylceramide + galactosylceramide) | HexCer d18:1 | 16-26 (+ OH 18-26) | d$_3$-glucosylceramide d18:1/16:0 |
| Dihexosylceramide (lactosylceramide + galabiosylceramide) | Hex2Cer d18:1 | 16-24 (+ OH 24) | d$_3$-glucosylceramide d18:1/16:0 |
| Globotriaosylceramide | Gb3Cer d18:1 | 16-18 | d$_3$-glucosylceramide d18:1/16:0 |
| Globoside | Gb4Cer d18:1 | 18 | d$_3$-glucosylceramide d18:1/16:0 |
| Globotriaosylsphingosine/lyso-Gb3 | Gb3Sph | n/a | N-glycinated globotriaosyl-sphingosine d18:1 |
| Phosphatidylcholine | PC | 34-40 | N-glycinated globotriaosyl-sphingosine d18:1 |
| Phosphatidylethanolamine | PE | 34-40 | N-glycinated globotriaosyl-sphingosine d18:1 |
| PE plasmalogen | PE(P) 16:0, 18:0, 18:1 | 18-22 | N-glycinated globotriaosyl-sphingosine d18:1 |
| Lyso-phosphatidylethanolamine | LPC | 16-22 | N-glycinated globotriaosyl-sphingosine d18:1 |
| Lyso-phosphatidylethanolamine | LPE | 16-22 | N-glycinated globotriaosyl-sphingosine d18:1 |
| Lyso-phosphatidylethanolamine PE | LPE(P) | 16-18 | N-glycinated globotriaosyl-sphingosine d18:1 |
| Sphingomyelin | SM | 34-36 | N-glycinated globotriaosyl-sphingosine d18:1 |
| Lyso-sphingomyelin | Lyso-SM | 16-18 | N-glycinated globotriaosyl-sphingosine d18:1 |
| Lyso-sphingomyelin 509/ N-palmitoyl-O-phosphocholineserine | Lyso-SM (509) | n/a | N-glycinated globotriaosyl-sphingosine d18:1 |
| Hexosylsphingosine /Lyso-Gb1 (glucosylsphingosine + galactosylsphingosine) | HexSph | n/a | d$_5$-glucosylsphingosine d18:1 |
| Dihexosylsphingosine/Lyso-Gb2 (lactosylsphingosine galabiosylsphingosine) | Hex2Sph | n/a | d$_5$-glucosylsphingosine d18:1 |

The table shows the analyte classes, abbreviations and monitored fatty acid chains, and the internal standards used to normalise the abundances. Throughout the manuscript, the term n-hexosylceramide is utilised to describe the collective group of ceramide with one to four hexosyl-groups. *d* deuterium-labelled, *LPC* Lyso-PC, *LPE* Lyso-PE, *LPE(P)* Lyso-PE(P), *LSM* Lyso-SM.

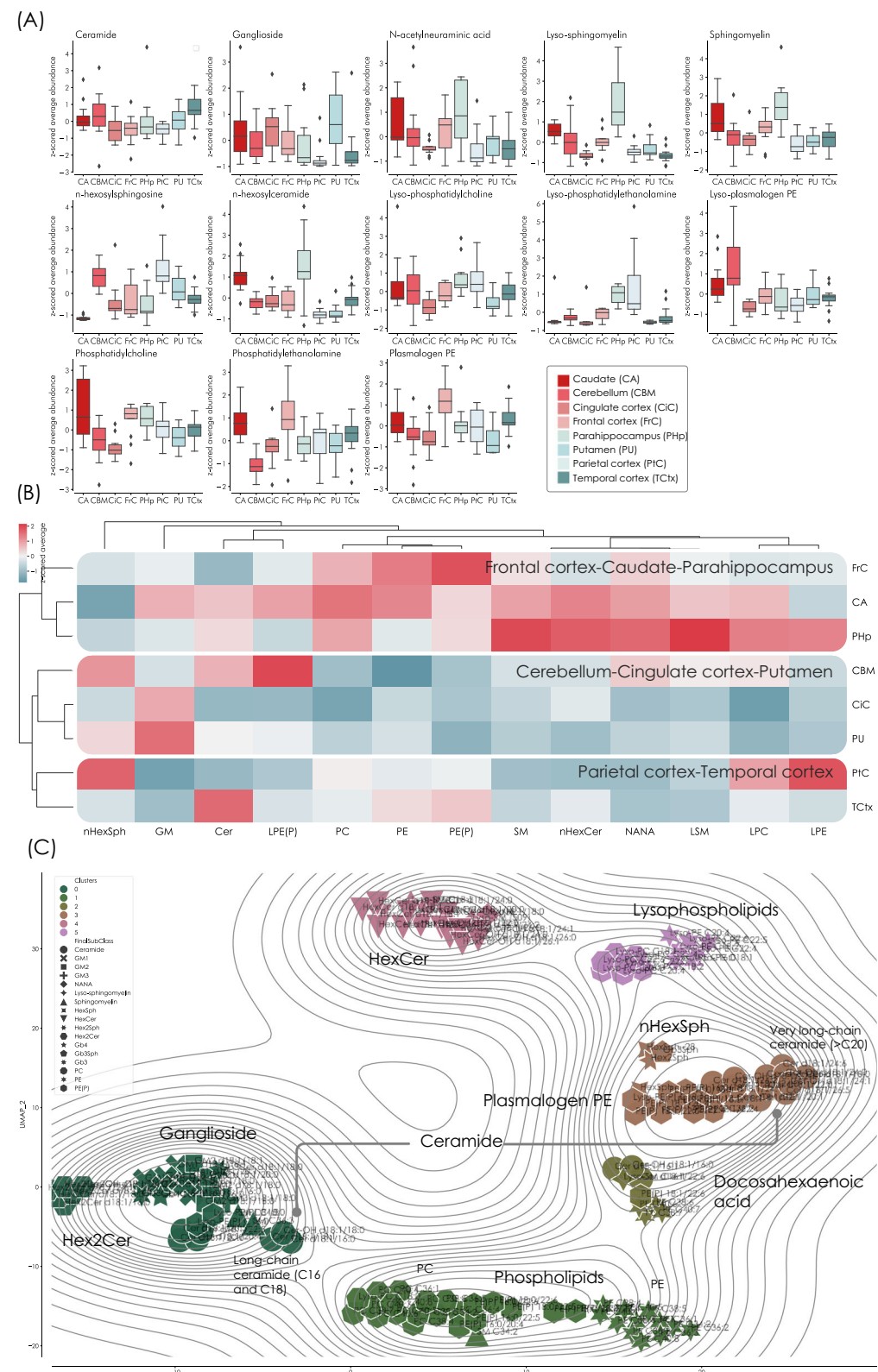

each other. We applied Hierarchical Density-Based Spatial Clustering of Applications with Noise (HDBSCAN) to cluster the lipids classes, identifying dense groupings in multi-dimensional feature space without requiring a pre-defined number of clusters. Six clusters emerged, generally confined of lipid species from the same class. We found that lyso-PE, lyso-PC and lyso-PE(P) formed a distinct cluster, and that PC and PE were projected in a general phospholipid cluster. Interestingly,

Hex2Cer grouped in the same cluster as ganglioside rather than HexCer, thereby indicating that HexCer levels are independent from the downstream lipids in the globoside branch of the glycosphingolipid degradation pathway. Also of note is that the ceramide species did not form one unique cluster but separated into two locations. The majority of ceramide clustered with plasmalogens, but also with the main ganglioside group. Generally, long-chain ceramide distributed near

**Fig. 1 | Lipid abundance in control brains without brain pathology. A** Box and whisker plots displaying the z-scored (represents the distance from the population mean in units of standard deviations) sum of lipids per class and the distribution in the different regions (CA $n = 13$, CBM $n = 14$, CiC $n = 13$, FrC $n = 12$, PHp $n = 11$, PtC $n = 15$, PU $n = 13$, TCtx $n = 16$). The whiskers show the minimum and maximum and the boxes show the 25th percentile, the median and the 75th percentile. Values outside 1.5 times the interquartile are represented by dots. It was demonstrated that the phospholipids, ceramide and sphingomyelin displayed an even distribution between the different regions. **B** Hierarchical clustering showing the average of each compound per region in controls. Three main clusters formed consisting of (i) frontal cortex, caudate and parahippocampus, (ii) cerebellum, cingulate cortex and putamen, and (iii) parietal and temporal cortices. The primary driving parameter for this clustering was found to be an elevated abundance of most lipids in the frontal cortex-caudate-parahippocampus cluster. The clustering method was set to average, with cosine as distance metric. **C** UMAP projection with HDBSCAN clustering, demonstrating how the lipid classes relate to each other. The projection grouped into six main clusters, largely made up of lipids from the same class although overlap between classes was also observed. The UMAP points were coloured according to main class and shaped according to sub-class. The projection was set to model 12 neighbours (average lipid class size) with correlation as the distance metric. Source data are provided as a Source Data file.

ganglioside, while very long-chain ceramide was found in the plasmalogen group. We additionally identified one cluster largely consisting of lipids with docosahexaenoic acid (DHA) fatty acids, mainly represented by PE and PE(P).

To determine whether age affects the brain lipid profile, we examined the control samples and noted that n-hexosylceramide was affected across all brain regions, apart from the cingulate and frontal cortices, and increased in abundance with increasing age (Supplementary Results Figure S2). We also evaluated sex-related variations in the lipid profiles by Orthogonal Partial Least Squares–Discriminant Analysis (OPLS-DA) but did not find any significant differences in abundance (ANOVA CV $p = 1$).

In conclusion, we found that the lipid profile differs regionally across the brain in controls and that age influences the lipid profile. The caudate and the temporal cortex were the regions most affected by age and were found to be particularly susceptible to age-related lipid changes, specifically in n-hexosylceramide and ceramide. Importantly we demonstrate that lipid subspecies, in particular ceramide, can behave independently from each other. These findings underscore the importance of comparing different regions when performing lipidomic analyses rather than using whole brain homogenate, as the regional lipidomes differ drastically and that care should be taken in reporting total lipid levels.

### Age adjustment

To avoid interpreting differences in age between the groups as relevant to discriminating between PD and controls, given the significant correlations between age and several of the lipids in the control group, we adjusted the data region-wise for age before performing any further analyses. Only the variables significantly correlated with age were adjusted and evaluated in both a total- and a region-wise OPLS model with age set as the y-vector to verify that the adjustment was effective.

None of these models was significant, thereby demonstrating that the age-adjustment was performed successfully. The $p$ values showing if a variable was significantly related to age are shown in Supplementary Results Table S1.

### Ceramide, sphingomyelin and ganglioside levels drive the changes observed in the global PD brain lipid profile

We next explored differences between PD and control. We constructed a PCA model of the PD and control samples throughout the brain regions. The first and second principal components largely separated the samples into controls and PD and the corresponding loadings showed that gangliosides were elevated in PD, along with long-chained and polyunsaturated phospholipids (Fig. 2A, B). The third and fourth principal components demonstrated that the samples formed groups according to which anatomical structure they belonged (basal ganglia, cerebral cortex, limbic system and cerebellum). Within the regional clusters, it was further possible to distinguish the separate brain regions (Supplementary Results Figure S3).

The PCA loadings demonstrated that the eight different PD brain regions were enriched in different lipid classes, again highlighting that different parts of the brain have different lipid profiles. In likeness to what we observed in the control samples alone, we noted that the cerebral cortex was abundant in ceramide, phospholipids and plasmalogen PE, while the inner brain regions were richer in ganglioside and lyso-n-hexosylsphingosine. The cerebellum demonstrated higher levels of lyso-lipids in general (both glycosphingo- and phospholipid species) (Fig. 2C, D). To evaluate the overall lipid profile related to the PD and control groups, we adjusted the data for brain regions before modelling the classes by discriminant OPLS-DA (Fig. 2E, F). This model was found to be highly significant (ANOVA CV $p = 3.1\,E^{-10}$) and no regional influence could be discerned. The predictive loadings showed that gangliosides GM1, GM2 and GM3 were elevated in PD, as were sphingomyelin and lyso-sphingomyelin. Lyso-phosphatidylethanolamine was lower in PD, while lyso-phosphatidylcholine was higher. Diacylated phospholipids and polyunsaturated plasmalogen PE were generally elevated in PD. The lipids most strongly responsible for the difference between the control and PD groups were higher levels of the ceramides with C16 and C18 fatty acid chains and lower levels of ceramide with very long-chain fatty acids (C20 and longer) in the PD group.

### Regional lipid alterations reflect neuroanatomical differences in Parkinson's disease

We further assessed the univariate and region-specific differences between PD and controls. We extracted signed (based on fold change direction) -log$_{10}$ $p$ values and built a hierarchical cluster to summarise the changes (Fig. 3A). The brain regions formed two main clusters, consisting of parahippocampus and the cingulate, temporal, frontal and parietal cortices in one, and the putamen, caudate and cerebellum in the other. These clusters (Fig. 3B) indicate that different molecular changes occur in PD between different regions and that these changes are related to the region's structure and/or function. In the first cluster, which consisted of regions from the cerebral cortex (frontal, temporal and parietal cortices) and the limbic system (cingulate cortex and parahippocampus), we found gangliosides and several HexCer and Hex2Cer species elevated in PD, while most phospholipids were lower. In the second cluster, consisting of the basal ganglia regions putamen and caudate, and of the cerebellum, certain phospholipids, particularly PC and lyso-PC, were higher in PD than control, while glycosphingolipids were generally found lower. These clusters may suggest neuroanatomical distinctions in the brain in Parkinson's disease and indicate shared and unique pathological signatures between the regions. The cortical cluster consists of structures involved in cognitive processing, memory, and sensorimotor integration and is generally affected in late stages of the disease[21,22]. The second cluster includes subcortical structures closely tied to PD pathology and is affected in early stages of PD[23,24]. The putamen and caudate are central to the basal ganglia and are heavily affected by dopamine depletion, leading to motor dysfunction. The cerebellum, though not traditionally considered a primary site in PD, has gained attention for its compensatory role in motor control[25–27].

We evaluated the total number of changed compounds throughout the different regions based on Benjamini-Hochberg multiple testing adjusted $p$ values (FDR = 10%) and noted that the parietal cortex was the region with the largest number of altered lipids

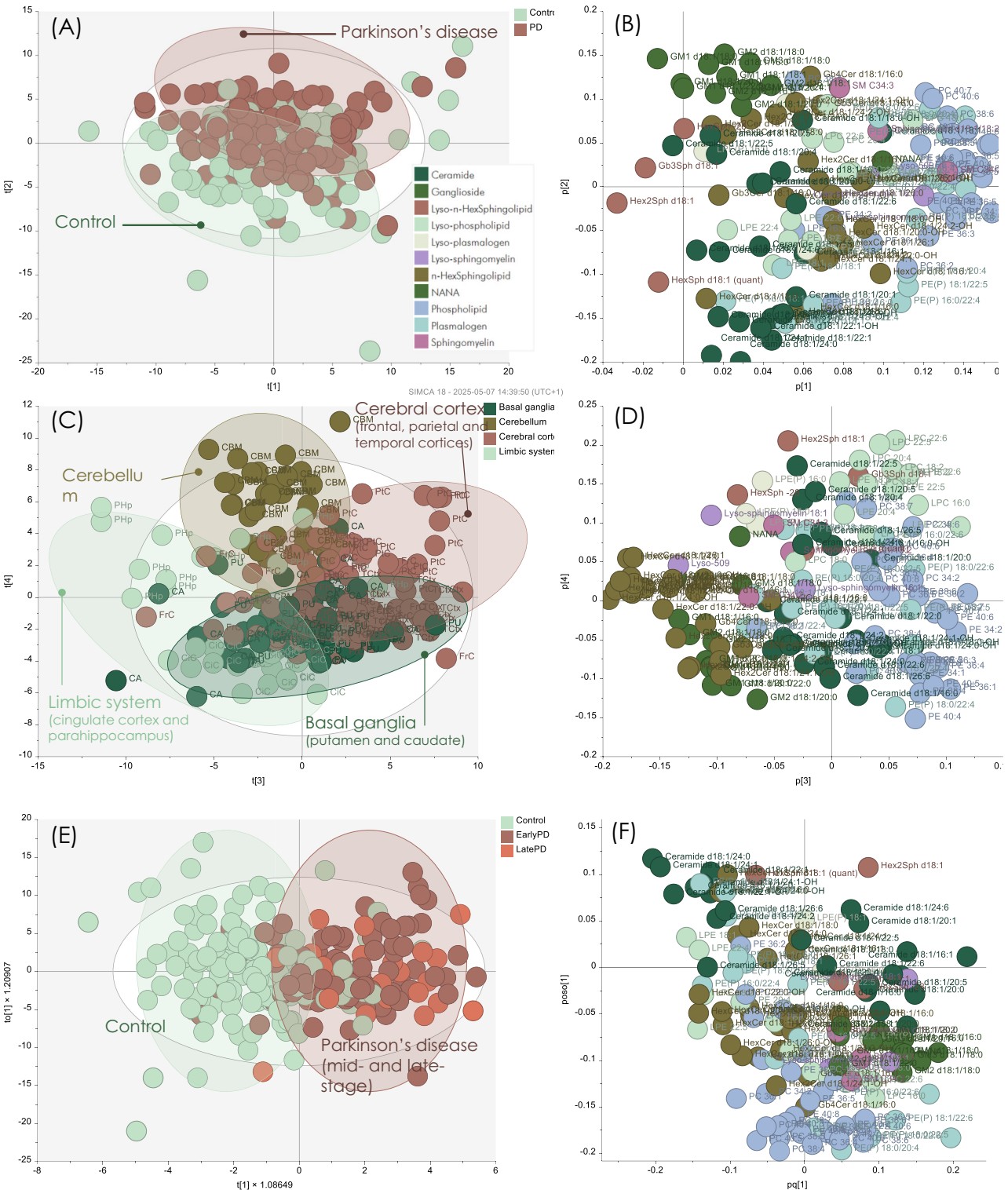

**Fig. 2 | Multivariate analysis of PD and control shows regional and disease-related differences. A** PCA scores, the first (20% model variation) and second (14% model variation) components largely separated the samples into PD and control. **B** The corresponding loadings showed that gangliosides were elevated in PD, along with long-chained and poly-unsaturated phospholipids. **C** PCA scores from the third (11%) and fourth (9%) components grouped the samples into anatomical brain regions, where the putamen and cingulate cortex overlapped with caudate and the frontal cortex. The other regions generally formed well-defined clusters. **D** The corresponding loadings demonstrated that the eight different PD brain regions

were enriched in different lipid classes. The cortices were rich in ceramide, phospholipid and plasmalogen, while the inner brain regions were richer in ganglioside and lyso-n-hexosylsphingosine. **E** The *cross*-validated scores from a region-adjusted OPLS-DA model of PD versus control, showing a clear separation between the groups. **F** Predictive OPLS-DA loadings showing the significant lipids where positive values represent lipids elevated in the PD group and negative values represent lower levels in PD. The model was significant with one-factor ANOVA CV $p = 3.1E^{-10}$ and permutations $p \ll 0.001$. Source data are provided as a Source Data file.

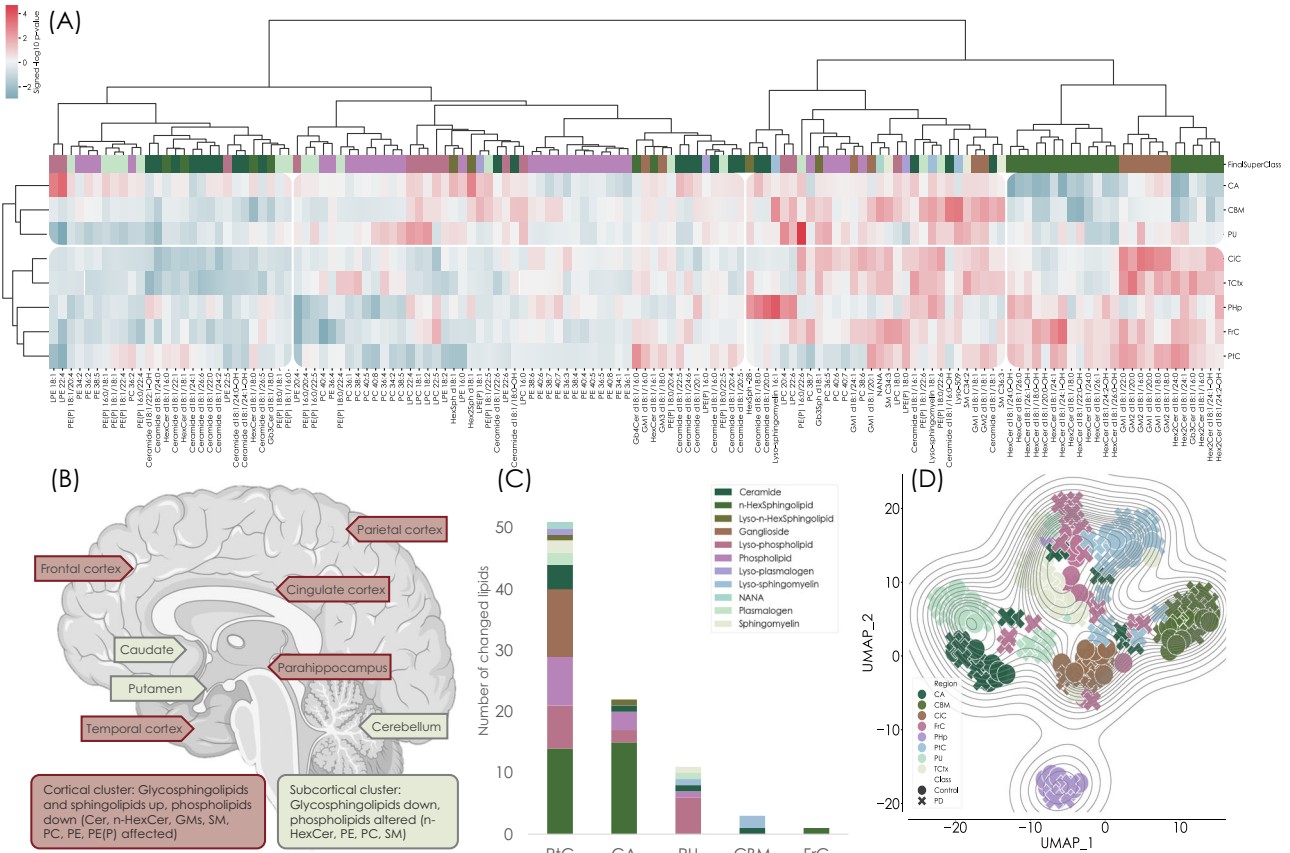

**Fig. 3 | There are region-specific differences in the lipid brain profiles between Parkinson's disease and controls. A** Hierarchical clustering of the profile differences (-log10 nominal *p* value) identified between control and PD in the eight different brain regions determined by two-sided Student's t-test or Mann-Whitney's U-test. Two main clusters formed consisting of (i) frontal, cingulate, parietal and temporal cortices and parahippocampus, and (ii) putamen, caudate and cerebellum. The clustering method was set to Ward, and the metric was Euclidean. Red represents an increase, and blue a decrease, in PD. **B** Differential expression of lipids across different regions of the brain. **C** Distribution of the differentially expressed lipids (Benjamini-Hochberg multiple testing corrected, FDR = 10%) in the brain regions, given as the total percentages of the nominally different lipids.

The region with most numerous changes was the parietal cortex (58% of the total changes), followed by caudate (25%) and the putamen (13%). **D** UMAP projection showing how the brain regions relate to each other. The regions generally formed distinct clusters separated from each other. Especially the cerebellum was distinctly different from the other regions. The difference's between the regions was found to be greater than the difference between PD and control samples. The UMAP was set to model 28 neighbours (average region sample size), and the metric was correlation. The lipid classes in (**A**) and (**B**) are both colour coded according to the legend in the lower left corner. *Part of figure* Created in BioRender. Baldwin, T. (https://BioRender.com/ s8g7xcr). *Source data are provided as a Source Data file.*

between PD and control (58% of the total lipids), followed by caudate (25%) and the putamen (13%). When assessing the compound classes, we found that a greater proportion of the sphingolipids were changed compared to the phospholipids. In the parietal cortex where the most numerous changes occurred, we identified equal numbers of phospho- and sphingolipids altered, while in the caudate, the majority consisted of glycosphingolipids (Fig. 3C).

We next projected the samples in a UMAP. We found that samples from the same region generally clustered together and that the basal ganglia regions were in close proximity to each other, as were the cerebral cortex regions, while the parahippocampus was found far from the other regions. The regional clustering emphasises that, although the regions have different lipid profiles, regions located spatially near each other also share similarities. No distinct separation could be identified between PD and control samples in the UMAP, signifying that the spatio-anatomical differences are greater than the differences between healthy brains and PD (Fig. 3D).

We identified region-specific pathological signatures that may contribute to disease progression. Lipid alterations in PD follow distinct neuroanatomical patterns, with cortical and subcortical regions exhibiting different molecular signatures. Cortical areas (para-hippocampus, cingulate, temporal, frontal, and parietal cortices),

affected in later disease stages, show elevations in gangliosides and glycosphingolipids, while phospholipid levels are generally reduced. In contrast, the cerebellum and the early-affected subcortical regions (putamen and caudate) display lower glycosphingolipid levels alongside increased phospholipids, particularly PC and lyso-PC.

### Region-specific lipid alterations align with disease progression in Parkinson's disease

Samples were classified according to Braak staging, defining mid-stage PD as Braak 3–4 and late-stage PD as Braak 5–6. Differences between the two PD progression stages and controls were evaluated in three regions with sufficient sample representation across both stages: the cerebellum, frontal cortex, and putamen (details in Supplementary Methods Table S1). According to Braak's framework of PD pathology progression, the putamen is the most affected region during mid-stage PD due to its role in the basal ganglia and dopaminergic signalling[28]. In contrast, the frontal cortex remains largely unaffected at this stage, but exhibits pronounced pathological changes in late-stage disease, consistent with cognitive and executive function impairment in PD. Meanwhile, the cerebellum, traditionally considered less involved in PD pathology, does not show overt pathological aggregation across disease progression, although emerging research suggests potential

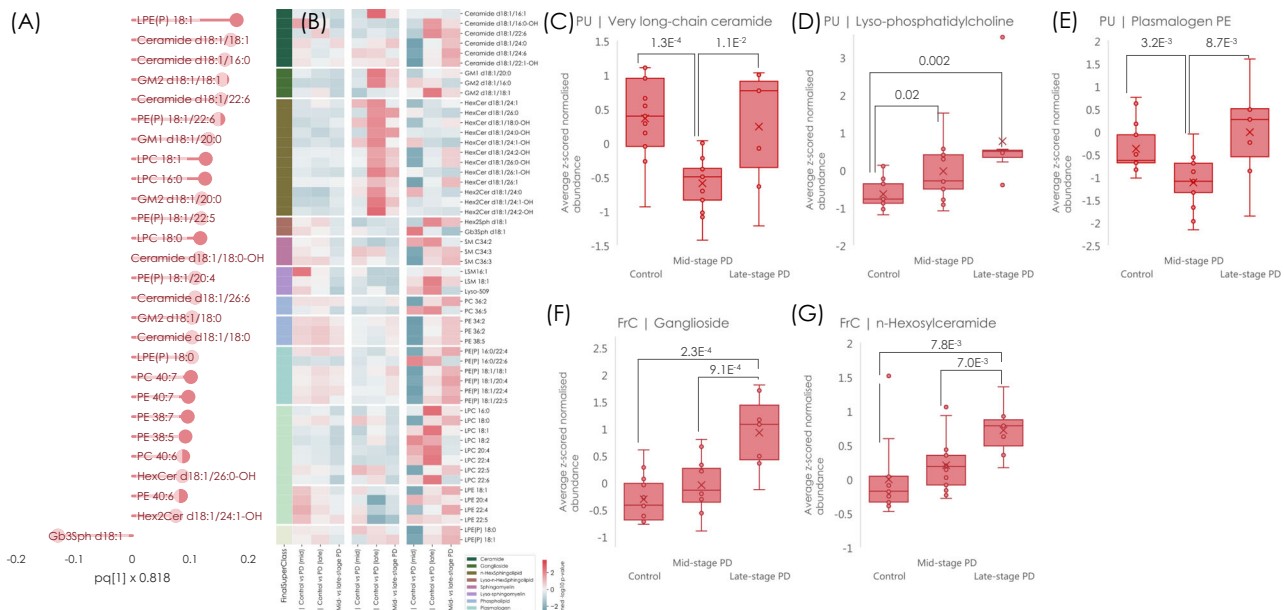

**Fig. 4 | Lipids in the cerebellum, frontal cortex and cerebellum in mid-stage and late-stage PD change with disease progression.** A OPLS-DA loadings from the significant (one-factor ANOVA CV $p = 0.005$) model of mid-stage versus late-stage PD, where positive values signify lipids elevated in the late stages of PD and negative values signify elevation in mid-stage PD. **B** Heatmap showing the Benjamini-Hochberg adjusted (FDR = 10%) -log10 $p$ values (two-sided Student's t-test for normally distributed variables, and Mann-Whitney's U-test for non-normally distributed variables) of the comparison between control, mid-stage and late-stage PD in the three regions cerebellum, frontal cortex and putamen. The sign was determined by fold change. **C–G** Box plots showing the average of the key altered lipids in mid- and late-stage PD, with (**C**) very long-chain ceramide species depleted in mid-stage PD ($n = 13$) in the putamen compared to controls ($n = 13$), **D** lyso-phosphatidylcholine elevated in both mid- and late-stage ($n = 7$) PD in the putamen region, **E** plasmalogen PE depleted in putamen in mid-stage PD, then normalising in late-stage PD, (**F**) ganglioside elevated in late-stage PD ($n = 7$) in the frontal cortex compared to controls ($n = 12$), and (**G**) n-hexosylceramide elevated in late-stage PD in the frontal cortex. The whiskers show the minimum and maximum and the boxes represent the 25th and 75th percentile, and the median. Significance levels in (**C–G**) are based on nominal and two-sided Student's t-tests. Source data are provided as a Source Data file.

compensatory mechanisms in motor control[29,30]. To investigate molecular differences, we first examined the global multivariate expression patterns and observed significant separations between controls and PD samples at both disease stages. When modelled via OPLS-DA, differences between controls and PD were highly significant ($p = 2.9E^{-6}$ in mid-stage PD, $p = 1.7E^{-6}$ in late-stage PD, Supplementary Results Figure S4). Importantly, a distinction between mid- and late-stage PD was identified ($p = 0.005$), largely driven by elevated levels of lysophospholipids, ceramide, ganglioside, and polyunsaturated diacylated phospholipids in late-stage PD samples (Fig. 4A).

Univariate and region-specific analyses aligned with Braak's staging, showing distinct lipid alterations across the brain regions (Fig. 4B). In the cerebellum, changes were limited, with only isolated elevations observed in ceramide and lyso-sphingomyelin in mid-stage PD. In the frontal cortex, lipid differences were restricted to late-stage PD, where increases in ceramide, ganglioside, hexosylceramide, and dihexosylceramide were observed, alongside lower levels of lyso-phosphatidylethanolamine. The putamen displayed extensive lipid alterations across both disease stages. Mid-stage PD was characterised by a widespread reduction in sphingolipids, particularly ceramide and HexCer species containing very long-chain fatty acids, suggesting early disruptions in lipid homoeostasis while, conversely, the shorter chain C18 ceramides were elevated (prominent in the region-adjusted OPLS-DA model, Fig. 4A) thereby indicating abundance of these ceramide species is unrelated to the longer >C20 ceramides. Several phospholipids were also elevated, a pattern that persisted into late-stage PD, where additional increases were noted in PC 36:1, PE(P) 16:0/22:5, PE(P) 18:1/22:6, and LPC 18:1, 18:2. Late-stage PD showed broad elevations across phospholipid and sphingolipid classes, with LPC species most prominently affected (Fig. 4C-G). These findings provide further

evidence of region-specific lipid alterations in PD, suggesting both functional and structural differences related to disease progression.

The lipid profile across Braak stages suggests a progression gradient in PD, with distinct molecular changes observed between mid-stage and late-stage PD. Lipid disruptions followed Braak staging, with minimal cerebellar changes, frontal cortex alterations emerging only in late-stage PD, and putamen exhibiting changes at both stages. Mid-stage PD was characterised by sphingolipid depletion, particularly ceramide and HexCer with very long-chain fatty acids, while late-stage PD showed increased levels of phospholipids and sphingolipids, including ganglioside, nHexCer, lyso-PC and plasmalogen PE.

## Multi-omic analysis correlates sphingomyelin-levels in the putamen with proteins altered in PD

We next evaluated the correlation between the protein abundances in the brain samples and the measured lipids, using Spearman correlation. Untargeted proteomics was used to profile these same samples, detailed in the publication by Toomey et al[31]. Significant correlations passing a Benjamini-Hochberg FDR threshold set to 5% and proteins correlating with at least two lipids from the same compound class were considered. The results showed that three sphingomyelins significantly correlated with the abundances of several proteins, positively with 14 proteins and negatively with 29 proteins, while in total, 25 of these proteins were nominally significant between PD and control in the proteomic data alone. We extracted the KEGG pathways associated with the lipid-correlated proteins and found that "Parkinson's Disease" was the strongest enriched pathway followed by several other pathways of neurodegenerative disorders (Fig. 5). Gene ontology (GO) suggested that biological processes related to mitochondrial energy metabolism were enriched.

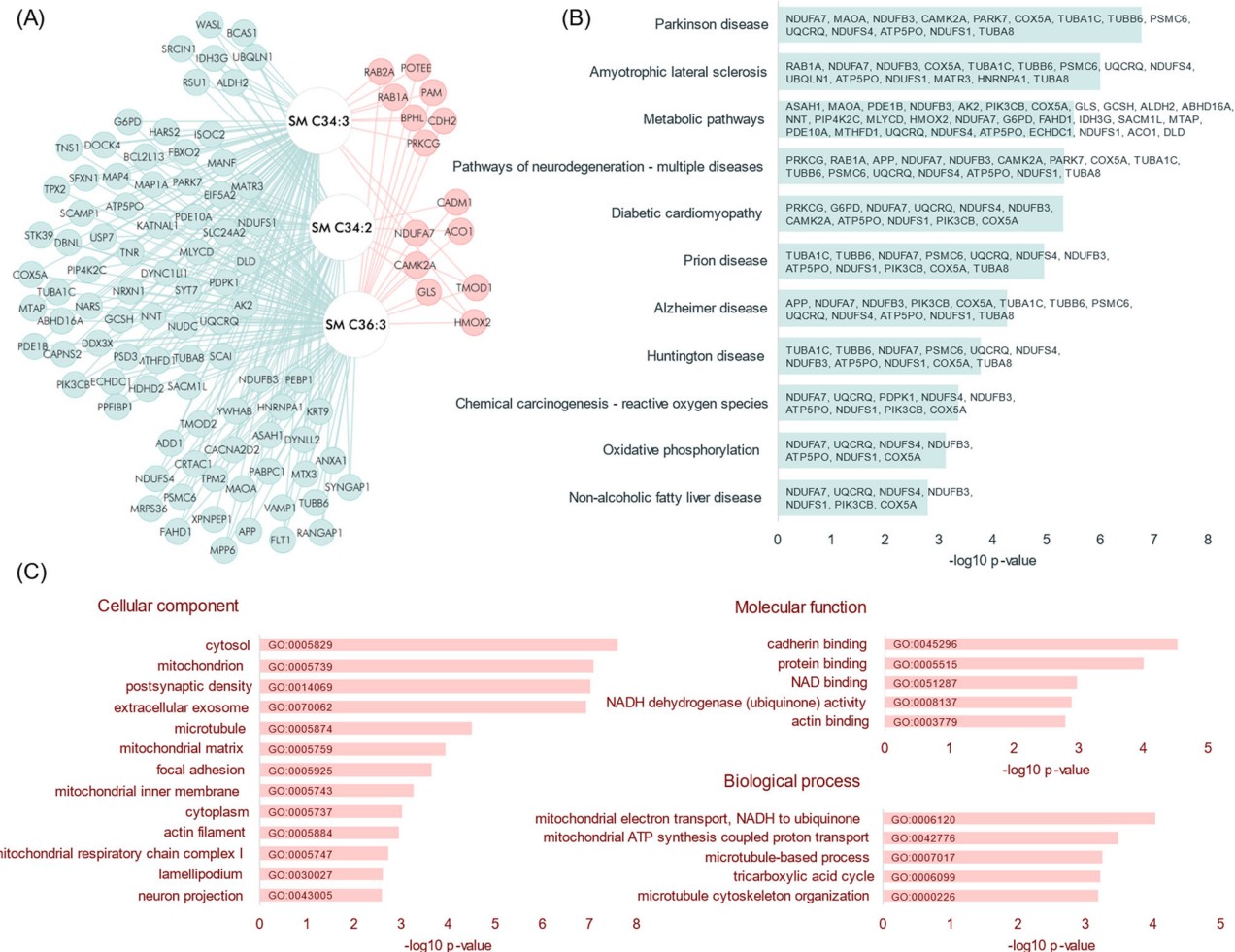

**Fig. 5 | Correlation between lipids and proteins.** The relationships between lipids and proteins in the regions putamen, parahippocampus and frontal cortex were evaluated by Spearman correlation, applying the Benjamini-Hochberg multiple testing correction procedure with FDR at 5%. The correlations were restricted to the ones significant in respective region post multiple testing correction, and only proteins which correlated with at least two lipids from the same compound class were considered. Only sphingomyelin in the putamen demonstrated significant differences between PD and controls, and significant lipid-protein correlations. **A** The network shows the significant interactions identified between proteins and sphingomyelin species in the putamen. Blue nodes represent negative correlations and red nodes positive correlations. Enriched pathways and GO annotations were extracted for significantly correlated proteins. The significantly enriched KEGG pathways (significance determined by two-sided Student's t-test and passing Benjamini-Hochberg FDR = 5%) are shown in (**B**) and demonstrate Parkinson's disease as the top hit together with several other neurodegenerative conditions. **C** shows the significantly enriched GO cellular component terms, and the top five GO molecular function and biological process terms. *Source data are provided as a Source Data file.*

The correlation of sphingomyelin with several PD-implicated proteins suggests that there exist disease-specific interactions between lipids and proteins and highlights the importance of bringing together multiple omics platforms to identify more specific correlations and changes related to disease.

**Lipids in the putamen show significant correlations with mitochondrial complex I activity**

We wanted to explore if any of the lipids were significantly correlated to markers of mitochondrial activity, given that the lipid-protein correlation-based pathway analysis suggested that mitochondrial energy activity was enriched. The mitochondrial complexes I, II-III, IV and citrate synthase were measured in the regions frontal cortex, parietal cortex and putamen as previously described[31] and it was found that all of the complex ratios I, II-III, and IV to citrate synthase were significantly different between the PD and control groups in the putamen region, with complex-ratios I and II-III lower in PD, and complex-ratio IV higher in PD (Supplementary Results Figure S5). The relationship between the lipids and the ratios of the mitochondrial complexes to citrate synthase were evaluated in a linear mixed

effects model where the brain regions were modelled as a random effect. The $p$ values from the interactions between the class variable (PD or control) and the complex proteins ratioed to citrate synthase were adjusted for multiple testing by the Benjamini-Hochberg procedure, and statistical significance accepted up to FDR = 10% (Supplementary Results Table S2). The results were visualised in a network (Fig. 6) where we found that several lipids, consisting of a large proportion of the compounds in the plasmalogen PE and lyso-PC classes, and also of ceramide, HexCer and HexSph, were significantly different between PD and control in the regression of complex I/CS levels to lipid levels. We only found one significant lipid related to complex II-III levels–ceramide C22:6, and none significantly different in complex IV.

The significant relationship between lipids and mitochondrial complex I suggests that these entities may modulate each other and would benefit from further investigation. We also measured the mitochondrial membrane lipid cardiolipin, but this was observed to be highly variable in all brain samples (42–52% CV in controls). We observed a lower trend in PD affected putamen tissue but likely due to the high level of variation, this was not significant.

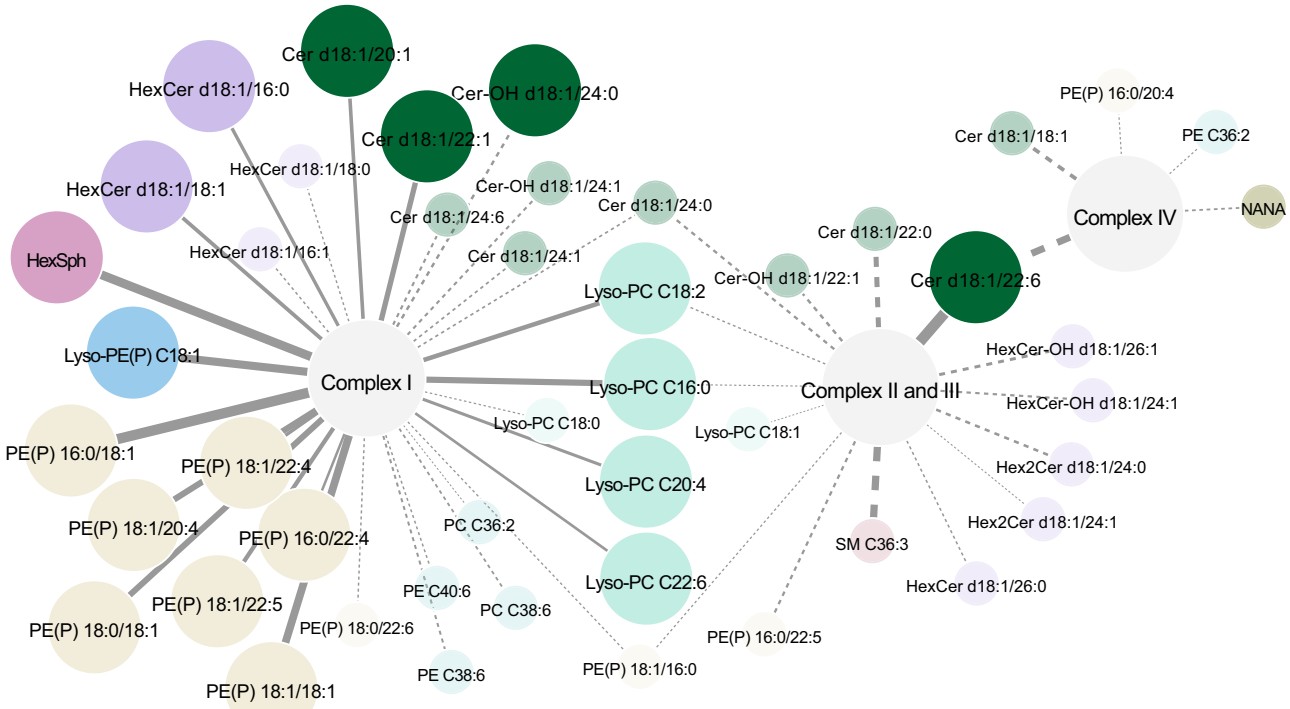

**Fig. 6 | Network of linear mixed model results, showing significant interactions between class (PD or control), and mitochondrial complex I, II-III, IV ratioed to citrate synthase.** The large circle nodes represent compounds statistically significant after Benjamini-Hochberg multiple testing correction at FDR = 10%, and the smaller radius nodes connected by dashed edges represent compounds significant on a nominal level. Source data are provided as a Source Data file.

## Discussion

In this study, we provide a comprehensive assessment of lipid alterations in the human brain in PD, highlighting both regional differences and disease progression-related changes. By examining lipidomic profiles across multiple brain regions and adjusting for age-related variation, we identified distinct lipid signatures associated with PD pathology.

Causation of sporadic PD has been found to be heterogenous for mitochondrial or lysosomal dysfunction[32]. This heterogeneity could confound the analysis; therefore, we examined the data for any subgrouping of lysosomal lipids to see if there was a potential lysosomal dysfunction signature, but we could not identify any evident subgroups among the PD samples, thus suggesting that the dataset could be treated as a whole. To ensure that observed lipid differences in PD were truly disease-related rather than arising from normal age-associated variation, we first established baseline lipid profiles in healthy, aged brains. This control cohort analysis provided key insights into regional lipid distributions, demonstrating distinct anatomical lipid signatures even in the absence of pathology. The most conserved lipids, which exhibited minimal variation across control brain regions, were ceramides. These had a coefficient of variation of 15% between brain regions, with a higher normalised abundance only observed in the temporal cortex. This minimal variation in ceramide levels across the brain suggests that their regulation is critical, and its homoeostasis is well controlled within the brain.

It is well-established that brain volume decreases with age and previous studies have highlighted the shrinkage of the hippocampus and cerebellum in particular[33] with the dentate gyrus of the hippocampus being the region most susceptible to ageing[34]. Our analysis of age-related changes demonstrated that age indeed has an evident effect on the brain lipid profile, with the putamen being most affected. Certain long-chain ceramide species, along with PC and lyso-PC, were strongly influenced by age, but it was n-hexosylceramide that exhibited the greatest overall change. These findings corroborate

observations from previous studies in humans and mice that have shown glucocerebrosidase activity progressively declining in the substantia nigra and the putamen in normal ageing, thus emphasising ageing as a risk factor for PD[35,36]. We explored if changes in PD mirrored those of a non-pathological advanced aged brain, but the only similarities we found were elevation of sphingomyelin C34:2 and C34:3, phosphatidylcholine C38:7 and lyso-SM 509 (an analogue of lyso-SM named *N*-palmitoyl-*O*-phosphocholine serine[37] [PPCS]).

Our study further reveals that lipid composition is highly region-specific, aligning with neuroanatomical distinctions and functional roles within the brain. The regions subjected to lipid profiling included the structures basal ganglia (putamen and caudate), the limbic system (parahippocampus and cingulate), the cerebral cortex (temporal, frontal, and parietal cortex), and the cerebellum. Each structure exhibited a distinct lipid profile, and within the structures, regional differences were discernible within them. Notably, the cerebral cortex was enriched in phospholipids and plasmalogen PE, whereas inner brain regions showed higher levels of ganglioside. The cerebellum, in contrast, displayed an overall abundance of lyso-lipids across both glycosphingolipid and phospholipid species. These results underscore the importance of investigating the brain's individual regions to fully appreciate the variations and the specificity of lipid metabolism in the brain, and its potential implications for neurodegenerative disease progression.

We also investigated how the lipids relate to each other in the brain. A key observation was that the ceramide subspecies do not correspond with each other as we identified two computationally generated clusters of different ceramides. One of these groups, rich in C16 and C18 ceramides, corresponded with ganglioside and, to a lesser degree, Hex2Cer whilst the other cluster, consisting of very long-chain species (>C20) grouped with plasmalogens. Ceramide has various other pathways feed into it such as the de novo synthesis, sphingo-myelin hydrolysis, sphingosine-1-phosphate and glycosphingolipid pathways[38]. These differences in abundances could be specific to the

different ceramide homoeostatic pathways. It is important to highlight that information can be missed or incorrectly reported on lipids by only reporting on total levels without looking at the subspecies' abundances. When examining the PD lipid profile, we observed a distinct ceramide profile which was characterised by chain-length-specific alterations. Long-chain ceramides (C16–C18) were elevated in PD in a global analysis and the most important variables separating between PD and control, whereas very long-chain ceramides (C20 and above) were reduced in the putamen. Referring to the control data analysis of lipid correlations, this corresponds to the long-chain ceramides clustering with ganglioside and the very long-chain ceramides (> C20) clustering with plasmalogens. Also to note is that the cluster of plasmalogens and > C20 ceramides contain the same lipid species as the ones found associated with mitochondrial complex I activity dysfunction. This difference in ceramides has been previously observed in the brains of PD patients, along with an increase in ceramide synthase 1 (CERS1), which is responsible for generating the C18 species[39]. The authors postulated that upregulation of CERS1 could be a downstream effect in response to stress in pathogenesis of PD however a recent sphingolipid tracer flux study using dual labelling with [13]C16-palmitate showed that C16–C18 ceramides largely originate from the de novo pathway and that this pathway is upregulated when GBA is inhibited by CBE treatment in a macrophage cell model[40]. This would suggest that imbalance in ceramides originates from dysregulation of the GSL degradation pathway as the cell attempts to maintain ceramide levels. This then begs the question - if it is crucial to keep ceramide levels consistent in the brain as this was the lipid class we observed the least variation in across regions—what would be the implications of elevated C16–C18 ceramide levels be in the brain?

Ceramide C18 has been observed to induce mitophagy by anchoring LC3B II auto-phagolysosomes to mitochondrial membranes[41]. Mitophagy induced by accumulated ceramide has been described as an alternative pathway initiated by the cell to overcome defective PINK1 related mitophagy in PINK1 mutation PD. This mechanism of mitophagy may result in reduced β-oxidation leading to defective mitochondria, thereby increasing the need for clearance and resulting in a vicious cycle[42]. Our data and that of Abbot et al[39] indicate that this alternative mechanism of mitophagy could be implicated in sporadic PD and requires further investigation.

Hierarchical clustering of the lipid data demonstrated that regions traditionally implicated in cognitive processing (parahippocampus, cingulate cortex, and frontal and temporal cortices) formed one distinct molecular cluster, whereas basal ganglia regions (putamen, caudate) and the cerebellum formed another. This division suggests that PD-related lipid alterations occur in patterns that reflect the brain's structural and functional organisation. The basal ganglia cluster exhibited elevated phospholipids and reduced glycosphingolipids, whereas the cortical cluster showed increased gangliosides and n-hexosylceramide, suggesting greater relevance of the glycosphingolipid pathway in these regions. These differences likely reflect neuroanatomical functions, with the basal ganglia experiencing early lipid disruptions linked to membrane remodelling and neuroinflammation. In contrast, the cortical cluster, affected later in PD, showed elevated gangliosides, potentially as part of a neuroprotective or adaptive response as ganglioside, GM1 in particular, has protective and neurorestorative properties in Parkinsonism[43,44] and has been trialled as a therapeutic in PD[45,46]. Importantly, we found that disease progression is mirrored in lipidomic changes across affected brain regions. We divided the PD samples into mid and late Braak stages in the three regions putamen, frontal cortex and cerebellum, and observed, in agreement with Braak staging, lipid alterations in the putamen in mid-stage PD, while the frontal cortex displayed significant lipidomic shifts only in late-stage disease. The cerebellum showed limited changes, supporting its compensatory rather than primary role in PD pathology. The increase in lyso-phospholipids and

polyunsaturated diacylated phospholipids in late-stage PD may indicate heightened membrane remodelling or neuroinflammation associated with advanced disease[47]. We moreover identified lipids who provided a ladder-like increase in response to disease progression, including lyso-PC and plasmalogen PE 16:0/22:5 and 18:1/22:6. Plasmalogens are important in membrane dynamics and act as potent antioxidants[48]. They have previously been reported reduced in lipid rafts of early PD patients, with total levels in the brain unaffected[19]. We observed elevated levels of plasmalogens in the putamen of mid- and late-stage PD. In similarity to the stipulated neuroprotective response observed in ganglioside in the frontal cortex, this increase of plasmalogen could be providing a protective antioxidant effect.

Multi-omic analysis revealed that sphingomyelin correlated with mitochondrial proteins, but not with mitochondrial complex activity. However, we found that > C20 chain ceramides were associated with complex I activity, along with C18 HexCer, C16–18 plasmalogens and lyso-PC. Other lipids also correlated with PD complex I activity, including plasmalogens and lyso-plasmalogen, which act as scavengers for reactive oxygen species. With this observation, their increase in PD brain tissue may be due to reactive oxygen species generated from complex I. Lyso-PC also correlated with complex I in PD brain tissue. This deacylated lipid is generated by phospholipase A2, an enzyme of which increased activity has been observed in the serum of PD patients[49]. Additionally, increased lyso-PC C16 and C18 have been identified previously in a rat model of mid-stage PD, where the authors suggest that the increase could be caused by inflammatory processes. In relation to mitochondria, lyso-PC has been found to mildly affect mitochondrial permeability but significantly affect mitochondrial function[50]. The effect of Lyso-PC on alpha synuclein appears controversial where it has been suggested to bind to alpha-synuclein, reducing its pathological accumulation[51]. Controversly, another recent study found lyso-PC promotes alpha-synuclein aggregation[52]. Whether lyso-PC is good or bad for alpha synuclein needs to be definitively determined but it is interesting to note in our analysis the cerebellum is the brain region with enriched lyso-phospholipids, including lyso-PC, and is the region least affected in PD so there may exist plausibility that it is protective against alpha-synuclein aggregation.

In conclusion, this study provides a comprehensive lipid profile of the normal human brain, detailing variations across anatomical regions, age groups, and disease states. Our findings offer a valuable resource for researchers investigating global lipid dynamics in age-related neurodegenerative disorders.

In the context of Parkinson's disease, we identify four key lipid alterations:

i.  Reduced very long-chain ceramides (> C20) in the putamen, suggesting early lipid dysregulation
ii. Increasing lyso-PC with disease progression
iii. Increased antioxidant lipids, including gangliosides and plasmalogens.
iv. Higher levels of sphingomyelin species (C34 and C36), correlating with proteomic changes.

These insights strengthen the understanding of lipid metabolism in PD pathology and underscore its role in disease progression. Future research should explore how these lipid shifts interact with protein networks and metabolic pathways, paving the way for novel therapeutic strategies.

## Methods
### Brain tissue samples
The samples included in this study have been described by Toomey et al[31]. In short, post-mortem brain tissue was obtained from the Neurological Tissue Bank, IDIBAPS-HC-Biobanc, Barcelona; Human Brain Tissue Bank, Budapest; UK Parkinson's Disease Society Tissue

Bank, Imperial College London; the London Neurodegenerative Diseases Brain Bank, Institute of Psychiatry, King's College, London; Netherlands Brain Bank, Amsterdam; and the Newcastle Brain Tissue Resource. Informed consent was given in all cases. Cases and controls were matched as close as possible for age and sex, and all had a post-mortem delay of less than 20 h. Control subjects did not have a diagnosis of neurological disease in life and at post-mortem displayed age-related changes only. All cases were assessed for alpha-synuclein pathology and rated according to Braak staging. PD cases were categorised as either Braak stage 3 or 4 (mid) or Braak stage 6 (late). Ethical approval for the study was obtained from the Local Research Ethics Committee of the National Hospital for Neurology and Neurosurgery.

In total, we analysed 225 post-mortem biopsies from the brain regions caudate, cerebellum, cingulate cortex, frontal cortex, parahippocampus, parietal cortex, putamen and temporal cortex. At least 50% of both the mid-stage PD and the control samples were sampled in at least six of the regions. Late PD was generally only sampled in the frontal cortex, cerebellum and putamen. The demographics of the cohort are described in Supplementary Methods Tables S1.

### Preparation of samples for targeted LC-MS/MS lipid analysis

The samples were prepared as described by Toomey et al[31]. In brief, the post-mortem brain tissue was homogenised in 50 mM ammonium bicarbonate buffer with 2% of the detergent ASB-14, and an amount of homogenate equivalent to 300 mg of total protein was precipitated with acetone. The lipid-containing supernatant was collected, and evaporated using a vacuum concentrator, after which the dry metabolite fraction was stored at -80 °C until further use. The lipid-containing fraction was reconstituted in 75 μL methanol containing the internal standards $d_3$-ceramide (Cayman chemicals, 24396), $d_3$-GM1 (Matreya, 2050), $d_3$-GM2 (Matreya, 2051), $d_3$-GM3 (Matreya, 2052), $d_5$-glucosylsphingosine (Avanti polar lipids, 860636), $d_3$-glucosylceramide (Matreya, 1533) and N-glycinated globotriaosylsphingosine (Matreya, 1530). While non-category specific internal standards were used, which may limit accuracy in absolute quantification across distinct lipid classes, this approach is acceptable for relative quantification. The samples were shaken on a rotational shaker at 1500 rpm for 15 minutes and sonicated for 15 minutes prior to centrifugation at 16,900 x $g$ for five minutes. 70 μL supernatant was transferred to glass micro vials.

### Analysis of lipids by targeted LC-MS/MS

In total, we quantified 146 species belonging to 11 main compound classes. The samples were analysed using a triple quadrupole mass spectrometer (Waters TQ-S) equipped with an ESI source, coupled to a quaternary Waters Acquity liquid chromatographic separation system. Detection was performed in multiple reaction monitoring mode (MRM), where the transitions from precursor ions to class-specific fragment ions were monitored. The MRM transitions were selected to represent characteristic fragmentation patterns for each lipid species[53,54]. The LC methods were tailored to achieve maximum separation between the compound classes. To ensure that adequate analytical setups were employed for each of the compound classes, we utilised three separate LC-MS/MS methods with different column chemistries (HILIC, Amide, or C8). Details about the chromatographic separation parameters can be found in Supplementary Methods Table S2 and details about the multiple reaction monitoring (MRM) transitions, MS settings and ionisation modes can be found in Supplementary Methods Table S3. The compound classes were normalised to internal standards according to Table 1. Example chromatograms of key lipids are displayed in Supplementary Methods Figure S1.

Cardiolipin analysis was performed as previously described. Briefly, the lyophilised samples were resuspended in 200ul of 1uM IS ((C14:0)4-CL) and 10ul injected onto an Acquity UPLC HSS T3 1.8uM 2.1 × 100 mm column. The mass spectrometer was a Waters TQ-XS. A gradient method was used for separation[55] and extracted ion chromatograms were integrated for the individual cardiolipin species. These peak areas were ratioed to the IS and the total CL present in each sample was calculated by the addition of all the individual CL species.

### Data integration

Data were acquired in MassLynx 4.2 and transformed into text files using the application MSConvert from the package ProteoWizard[56]. Peak picking and integrations were performed using an in-house application written in Python (available via the GitHub repository https://github.com/jchallqvist/mrmIntegrate) which rendered area under the curve by the trapezoidal integration method. Each analyte was thereafter normalised to an internal standard as described in Table 1.

### Statistical methods and visualisation

If not otherwise specified, all statistical analyses were performed in Python (version 3.11.11). The dataset was inspected for outliers and instrumental drift using principal component analysis (PCA) and orthogonal projection to latent variables (OPLS) in SIMCA, version 18 (Umetrics Sartorius Stedim, Umeå, Sweden). Outliers exceeding ten median absolute deviations from each variable's median were excluded. Adjustment for age and regions was performed using multiple linear regression from Statsmodels (version 0.13.5). The data were evaluated for normal distribution using D'Angostino and Pearson's method from SciPy (version 1.9.3). Significance testing between the independent groups of control and PD, and of control, mid-stage PD and late-stage PD, was performed by Student's two-tailed t-test for normally distributed variables and by Mann-Whitney's non-parametric U-test, both from the SciPy's stats package, for the non-normally distributed variables. The Benjamini-Hochberg FDR discovery rate procedure (Statsmodels version 0.14.0) was applied with alpha set to 0.05 or 0.1. Fold-changes were calculated by dividing the means or medians of the affected groups by the control group. Correlation analyses in the targeted data were performed by Spearman's correlation (SciPy) and the correlation $p$ values were adjusted variable-wise by the Benjamini-Hochberg FDR procedure. UMAP projections were performed using the Python package Umap (version 0.5.3)[20]. HDBSCAN was performed using default settings. Box plots and hierarchical clusters were produced using the Seaborn (version 0.12.2) and Matplotlib (version 3.7.0) libraries. Linear mixed models were performed using the R-to-Python bridge software pymer4[57] (version 0.8.0), where region was set as a random effect and the interaction between the classes PD/control and the mitochondrial complexes I, II–III and IV were evaluated for significance post Benjamini-Hochberg's procedure[58] for multiple testing correction.

All multivariate analyses were performed in SIMCA. OPLS and OPLS-discriminant analysis (OPLS-DA) models were evaluated for significance by ANOVA $p$ values and by permutation tests applying 1000 permutations, where $p < 0.05$ and $p < 0.001$ were deemed significant, respectively.

Data were analysed for pathway enrichment and Gene Ontology (GO) annotations using DAVID Bioinformatics Resources (2021 build). Networks were built in Cytoscape (version 3.8.0) applying the "Organic layout" from yFiles.

### Reporting summary

Further information on research design is available in the Nature Portfolio Reporting Summary linked to this article.

## Data availability

The raw targeted chromatograms of the lipid data are available to view and download via the Panorama repository (https://panoramaweb.org/ PD_Brain_Lipids.url). Source data are provided with this paper.

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

## Acknowledgements

The authors gratefully acknowledge support and funding from the MRC and NIHR through the Lipidomics and Metabolomics for Rare Disease node, part of the UK Rare Disease Research Platform (award MR/Y008057/1), Michael J Fox Foundation, Parkinson's UK, the Alexander Peto Foundation, the Translational Mass Spectrometry Research Group at UCL, Ms Sara Guttman for preparation of additional samples, Eisai the Lila Reta Weston Trust and Aligning Science Across Parkinson's. This work is supported by the NIHR GOSH BRC. The views expressed are those of the authors and not necessarily those of the NHS, the NIHR or the Department of Health. RP is supported by the UK Dementia Research Institute (award number MC_PC_17114), which receives its funding from UK Dementia Research Institute Ltd., funded by the UK Medical Research Council (MRC), Alzheimer's Society, and Alzheimer's Research UK.

## Author contributions

J.H.: Conceptualisation, Data acquisition, Data curation, Formal analysis, Methodology, Writing—original draft. C.E.T.: Data acquisition, Data curation, Formal analysis, Writing—review & editing. R.P.: Formal Analysis, Methodology, Writing—review & editing. A.W., I.D., J.L., J.R.E., M.A.S., T.B.: Data acquisition, Formal analysis SE, TB SP, SE, SH: Supervision—review & editing. KM, SG and WEH: Conceptualisation, Supervision, Methodology, Writing —original draft.

## Competing interests

None of the authors have any competing interests to report.

## Additional information

¹Translational Mass Spectrometry Research Group, Genetic & Genomic Medicine, UCL Great Ormond Street Institute of Child Health, London, UK. ²Queen Square Brain Bank for Neurological Disorders, UCL Queen Square Institute of Neurology, London, UK. ³Department of Clinical and Movement Neurosciences, UCL Queen Square Institute of Neurology, London, UK. ⁴The Francis Crick Institute, London, UK. ⁵MRC Centre for Environment and Health, School of Public Health, Department of Epidemiology and Biostatistics, Faculty of Medicine, Imperial College London, London, UK. ⁶UK Dementia Research Institute at Imperial College London, London, UK. ⁷MRC-NIHR BRC National Phenome Centre, Section of Bioanalytical Chemistry, Division of Systems Medicine, Department of

Metabolism, Digestion and Reproduction, Faculty of Medicine, Imperial College London, Hammersmith Hospital Campus, London, UK. [8]Neurometabolic Unit, National Hospital for Neurology and Neurosurgery, Queen Square & UCL Great Ormond Street Institute of Child Health, London, UK. [9]College of Applied Medical Sciences, King Khalid University, Abha, Saudi Arabia. [10]Developmental Biology and Cancer University College London Great Ormond Street Institute of Child Health, London, UK. ✉e-mail: wendy.heywood@ucl.ac.uk

