## [Transparent Peer Review file · Nature Communications]

Multi-Omic Analysis Reveals Lipid Dysregulation Associated with Mitochondrial Dysfunction in Parkinson's Disease Brain

Corresponding Author: Dr Wendy Heywood

Version 0:

Reviewer comments:

Reviewer #1

(Remarks to the Author)

The article "Multi-Omic Analysis Reveals Lipid Dysregulation Associated with Mitochondrial Dysfunction in Parkinson's Disease Brain" (NCOMMS-24-11914-T) adds unbiased lipidomics data to previously published proteomics data from the same group (Toomey et al 2022 Acta Neurol Comm). The authors studied several phospho- and sphingolipid classes mainly in four distinct cortical regions and cerebellum, but also in putamen, caudate, parahippocampus of the human control and PD brains from different brain banks. The number of biopsies varied between regions and were only 5 for the PD group in the latter two regions. Targeted LC-MS/MS was applied and the obtained lipid data was correlated with previously published untargeted proteomics and mitochondrial activity data. It is concluded that the lipid profiles differ between brain regions, correlate with age, mitochondrial activity and that ganglioside, sphingomyelin and n-hexosylceramides are region-specifically altered in PD. The study appears technically sound, but major weaknesses are that it is mainly descriptive, underpowered and reanalyze already published data.

Specific points:

1. The demographics needs to be expanded. It is unclear how many specific subjects that are included in the study. In other words, how many subjects contribute with several brains regions? Indicate the number of subjects from each of the seven brain banks. Information on post-mortem interval before autopsy as well as medication in PD patients should be added in Table S1.
2. Since few samples from many brain banks are analyzed, the data should not only be corrected for sex and age, but also brain bank.
3. It is unclear how many subjects that are included in this study, but since GBA mutations are frequent and could have strong influence on lipid profiles, it would be interesting to know about the GBA genotype of the studied subjects.
4. It would be fairer to call Braak 3-4, mid-stages rather than early-stages as done in this manuscript.
5. Only 1 sample for late Braak stage (5-6) is analyzed for the majority of regions (Caudate, Cingulate cortex, parahippocampus, parietal cortex and temporal cortex), making all assumptions on correlations with Braak staging extremely speculative. Moreover, the concluding statement of the abstract "we identified a gradient corresponding to Braak's disease spread across the brain regions, where the areas closer to the brainstem/substantia nigra showed alterations in PC, LPC and glycosphingolipids, while the cortical regions showed changes in glycosphingolipids, specifically gangliosides, HexCer and Hex2Cer." Is difficult to understand as none of the studied brain regions is in close vicinity to the brainstem/substantia nigra.
6. Lipid expression profiles clusters among regions in Figure 1. However, except for parietal and temporal cortex, the reader has difficulties to see the connectoms and other (eg neurotransmitter) relations between them. How does frontal cortex, putamen ("motor striatum") and cerebellum make a physiologically meaningful cluster? Are there other factors (eg brain bank) that largely determines these lipid expression clusters in this study?
7. Related to point 5, it is contradictory that there are no lipid changes in putamen in Braak stage 3-4 biopsies, but only in 5-

6. Meanwhile there are changes in frontal cortex and cerebellum in Braak stages 3-4 biopsies.

8. Since the authors states in the title that "Lipid Dysregulation is Associated with Mitochondrial Dysfunction in Parkinson's Disease Brain", it would have been appropriate to study mitochondrial-enriched lipids, such as cardiolipins. Such analyses would provide some depth of this study.

9. The reviewer has difficulties to identify mechanistic conclusions from this work which would lead to novel insight in PD pathophysiology. For example, this study contradicts previous highly cited work related to ganglioside levels in brain samples from PD patients.

10. Pagination of the manuscript would help the reader

Reviewer #2

(Remarks to the Author)

This manuscript contains a wealth of information on how the brain differs in people with PD and how brain shows region specific differences. The authors focus on lipids, particularly sphingolipids and glycosphingolipids which show PD related differences. This is probably the most detailed study of lipids in PD brain to date.

I have a few questions and comments that the authors may like to consider.

1. Introduction. The authors state "the brain consists of 60-70% lipids". Is this % dry weight or mole %?
2. The term "isoform" is usually used in proteomics, perhaps its meaning in the current context could be defined.
3. Similarly, "expression" is a word better used in a proteomic context, here I think the authors are referring to lipid concentration or abundance.
4. Methods section. Table 1. While many of the internal standards are fine, in some cases they are not ideal e.g. for phosphatidylcholines. I think this is acceptable, as only relative concentrations are being reported, but some justification should be given.
5. I would have liked to see some representative chromatograms to get a better feels of the data. While quite a lot of lipids are reported, the number is not too overwhelming for representative chromatograms to be included in the supplementary.
6. It is very good to show all the MRMs in the supplementary, but perhaps some references could be included to help those not expert in shingolipid and phospholipid mass spectrometry to understand the transitions.
7. In Figure 1, it might help the reader if z-score was defined.
8. Throughout the manuscript the term "n-hexosylceramide" is used, what does the "n" signify?

The statistics were beyond my expertise, but clearly there is a lot of valuable information here.

Version 1:

Reviewer comments:

Reviewer #1

(Remarks to the Author)

It is appreciated that the authors now directly compared lipid data from two different brain banks to substantiate text that there are no major differences in tissue handling between these two brain banks.

It is also commendable that the authors studied the levels of cardiolipins as Parkinson related changes would have provided a mechanistic link between lipids and mitochondria as stated in the title of this manuscript. Unfortunately, the data did not show any disease related changes.

Based on the major concerns below, the reviewer thinks that this manuscript is better suited in a more specialized journal.

Major concerns

A major concern is still the very low number of brain samples analyzed. The authors have not added any samples in the revision process. There is only 1 sample for Braak stage 5-6 in Temporal cortex, parietal cortex, parahippocampus, cingulate cortex and caudate nucleus. The reviewer think that it is not possible to relate the data to stages of Parkinson's disease as done by the authors for example in figure 3. The authors only have data to compare controls with PD.

In the "mid stage cases", where there are more brain samples studied, most lipid changes occur in cerebellum. Despite that the authors cite some articles on a role of cerebellum in Parkinson's disease, it is a region considered to have a minor influence of the clinical symptoms and signs of Parkinson's disease. It is difficult for the reviewer to see that these data bring any substantial new insight into the pathogenesis of Parkinson's disease.

Reviewer #2

(Remarks to the Author)

This is a very interesting article showing how the phospholipid/sphingolipid profiles of human brain vary with location, age and disease state. The data will be of interest to clinicians, neuroscientists, biologists, biochemists and lipidologists. The variation of sphingolipid/phospholipid profiles in the different states of disease is fascinating, as is their clustering in different regions. The variation of lipid profile with Braak staging with further strengthens the manuscript. I only have minor comments and suggestions.

1. "Isoform" is not a word normally used in lipidomics I would recommend the authors use more standard nomenclature based on category (e.g. shingolipids), class (e.g. cermaides, phosphoshingolipids), sub-class (e.g. sphingomyelins). See J. Lipid Res. 2005.46:839–861.
2. While it is excellent practice to deposit data in the Panorama Repository, it would be nice to see some real chromatograms in the Supplementary Results.
3. The numbering of the Supplementary Figures seems to be out of sync with the main text.
4. Methods. The authors could add to the text the solvent the brain tissue was homogenized in.
5. Methods. I suggest a few words could be added explaining the limitations of using non-category specific standards. As relative quantifications are used I don't think their use materially weakens the manuscript.
6. What is the y-axis in Figure 4.

Besides these minor suggestions/comments, I see this as an excellent manuscript.

Version 2:

Reviewer comments:

Reviewer #1

(Remarks to the Author)

I would like to commend the authors for now performing a major revision of their manuscript adding a lot of data. This significantly enhanced its impact. I have no more concerns.

Reviewer #2

(Remarks to the Author)

The authors have made a satisfactory reply to the reviewers comments. I have only very minor comments remaining:
Line 99 could be rephrased for clarity.

NCOMMS-24-11914A Multi-Omic Analysis Reveals Lipid Dysregulation Associated with Mitochondrial Dysfunction in Parkinson's Disease Brain

Authors' response to Reviewers comments

Please note our responses are in blue italic after each reviewer comment

REVIEWER COMMENTS

Reviewer #1 (Remarks to the Author):

The article "Multi-Omic Analysis Reveals Lipid Dysregulation Associated with Mitochondrial Dysfunction in Parkinson's Disease Brain" (NCOMMS-24-11914-T) adds unbiased lipidomics data to previously published proteomics data from the same group (Toomey et al 2022 Acta Neurol Comm). The authors studied several phospho- and sphingolipid classes mainly in four distinct cortical regions and cerebellum, but also in putamen, caudate, parahippocampus of the human control and PD brains from different brain banks. The number of biopsies varied between regions and were only 5 for the PD group in the latter two regions. Targeted LC-MS/MS was applied and the obtained lipid data was correlated with previously published untargeted proteomics and mitochondrial activity data. It is concluded that the lipid profiles differ between brain regions, correlate with age, mitochondrial activity and that ganglioside, sphingomyelin and n-hexosylceramides are region-specifically altered in PD. The study appears technically sound, but major weaknesses are that it is mainly descriptive, underpowered and reanalyze already published data.

We would like to thank the reviewer for taking the time to evaluate this manuscript. We appreciate that our study has a limited number of samples, which were carefully sourced and handled. Regarding the reviewer's concern about previously published data, our manuscript presents original and previously unpublished lipid data, and the use of proteomic and mitochondrial activity data are to demonstrate the relevance of our novel lipid findings. These data, in conjunction with the novel lipid, data helped us to gain further insights into potential disease mechanisms which the lipid data would not have been able to achieve on its own. Therefore, whilst the proteomic data may already be published, it has been applied in a novel way. This is similar to how many public and pre-published datasets available on public repositories have been used in many other published studies.

Specific points:

1. The demographics needs to be expanded. It is unclear how many specific subjects that are included in the study. In other words, how many subjects contribute with several brains regions? Indicate the number of subjects from each of the seven brain banks. Information on post-mortem interval before autopsy as well as medication in PD patients should be added in Table S1.

The requested information has been included as supplementary methods table S3 and has been taken from the demographics information for these same samples provided in Toomey, C.E., Heywood, W.E., Evans, J.R. et al. Mitochondrial dysfunction is a key pathological driver of early stage Parkinson's. acta neuropathol commun 10, 134 (2022). <https://doi.org/10.1186/s40478-022-01424-6>

2. Since few samples from many brain banks are analyzed, the data should not only be corrected for sex and age, but also brain bank.

We include the sex and age differences we observed in the manuscript in the first section of results and document these. In regards to brain bank comparison, the way the samples were distributed between PD and controls across the brain banks made it unfeasible to adjust the data in this way. Samples at all brain banks are collected according to standardised protocols to minimise centre-to-centre variation and are stored at -80 °C within 24 hrs post-mortem. However, to reassure the reviewer and to ascertain that there are indeed no lipid concentration differences related to brain bank site, we have sourced further samples from two different brain banks and analysed these for lipids to determine if different brain banks could influence lipid concentrations. We compared four controls from a Netherlands brain bank with six controls from a Newcastle brain bank for the caudate, frontal and parietal cortices and putamen regions. We have provided the p values for each lipid in Supplementary data table S7 which show no significant differences between these two brain banks.

3. It is unclear how many subjects that are included in this study, but since GBA mutations are frequent and could have strong influence on lipid profiles, it would be interesting to know about the GBA genotype of the studied subjects.

These samples were selected as known sporadic PD cases and were negative for known PD associated mutations.

4. It would be fairer to call Braak 3-4, mid-stages rather than early-stages as done in this manuscript.

We have corrected early to mid stage throughout the manuscript and figures.

5. Only 1 sample for late Braak stage (5-6) is analyzed for the majority of regions (Caudate, Cingulate cortex, parahippocampus, parietal cortex and temporal cortex), making all assumptions on correlations with Braak staging extremely speculative. Moreover, the concluding statement of the abstract “we identified a gradient corresponding to Braak’s disease spread across the brain regions, where the areas closer to the brainstem/substantia nigra showed alterations in PC, LPC and glycosphingolipids, while the cortical regions showed changes in glycosphingolipids, specifically gangliosides, HexCer and Hex2Cer.” Is difficult to understand as none of the studied brain regions is in close vicinity to the brainstem/substantia nigra.

We have corrected and re-phased this last sentence in the abstract to be clearer.

We observed 2 clusters or patterns of lipid change across the brain regions in response to PD, one pattern where the regions in closer proximity to the brainstem/substantia nigra (caudate, putamen

and cerebellum) showed alterations in PC, LPC and glycosphingolipids, and another pattern showed changes in glycosphingolipids, specifically gangliosides, HexCer and Hex2Cer in the further distal cortical regions (frontal, temporal and parietal cortices).

6. Lipid expression profiles clusters among regions in Figure 1. However, except for parietal and temporal cortex, the reader has difficulties to see the connectoms and other (eg neurotransmitter) relations between them. How does frontal cortex, putamen (“motor striatum”) and cerebellum make a physiologically meaningful cluster? Are there other factors (eg brain bank) that largely determines these lipid expression clusters in this study?

The similarities between regions we observe in figure 1 is an unbiased and data driven observation of the lipid profile in the brain without bias from other known factors, such as known functions, physical proximity, etc. We do not know the physiological reason for these lipid profiles to cluster together but speculate that they are likely related to the specific lipid function in these regions, such as cell type membrane composition or biochemical processes and warrants further investigation. We have clarified in the results section that an unbiased and data driven approach was applied. As mentioned in point 2, we have additionally assessed whether different brain banks induce detectable changes in the lipid concentrations, and conclude that the brain bank site has no effect on the lipid profile.

7. Related to point 5, it is contradictory that there are no lipid changes in putamen in Braak stage 3-4 biopsies, but only in 5-6. Meanwhile there are changes in frontal cortex and cerebellum in Braak stages 3-4 biopsies.

We agree that this is not expected and we cannot provide a certain explanation, but we observed the same effect in the proteomic data published on these samples in Toomey et al 2022. This supports that the observed differences are indeed related to physiological changes and not a technical reason. We can only speculate this might be due to certain functions or vulnerabilities specific to those regions. As we mention in the discussion, the increased gangliosides - especially GM1 - is possibly part of a protective mechanism and released at the first sign of stress and/or dysfunction and may explain why these regions that are unaffected Braak stage wise are showing lipid changes.

8. Since the authors states in the title that “Lipid Dysregulation is Associated with Mitochondrial Dysfunction in Parkinson’s Disease Brain”, it would have been appropriate to study mitochondrial-enriched lipids, such as cardiolipins. Such analyses would provide some depth of this study.

We performed further lipid analysis for cardiolipin which was observed to be highly variable in control and PD brain tissue. Due to this high variation we could not observe a significant difference between control and PD. We have provided this analysis in supplementary data figure S4 and updated the manuscript with this observation.

9. The reviewer has difficulties to identify mechanistic conclusions from this work which would lead to novel insight in PD pathophysiology. For example, this study contradicts previous highly cited work related to ganglioside levels in brain samples from PD patients.

We thank the reviewer for this comment. We have addressed this conflicting observation in the discussion and note that previous reports have described reduced ganglioside levels in the

substantia nigra, something which we could not confirm due to a lack of material caused by disease degeneration. However, our data does corroborate the increased ganglioside levels seen in the other regions, as described by Blumenreich et al 2022. To our knowledge, Blumenreich's and our study are the only studies that have deeply profiled human PD brain for lipids in this way.

10. Pagination of the manuscript would help the reader

We have now paginated the manuscript. Apologies for this oversight.

Reviewer #2 (Remarks to the Author):

This manuscript contains a wealth of information on how the brain differs in people with PD and how brain shows region specific differences. The authors focus on lipids, particularly sphingolipids and glycosphingolipids which show PD related differences. This is probably the most detailed study of lipids in PD brain to date.

I have a few questions and comments that the authors may like to consider.

1. Introduction. The authors state "the brain consists of 60-70% lipids". Is this % dry weight or mole %?

This refers to dry weight which we have now updated in the manuscript

2. The term "isoform" is usually used in proteomics, perhaps its meaning in the current context could be defined.

For lipids the isoform term relates to the same species of lipid but with different modifications such as the number of carbons, fatty acid chains, hydroxylation and other possible modifications. We have clarified this in the introduction.

3. Similarly, "expression" is a word better used in a proteomic context, here I think the authors are referring to lipid concentration or abundance.

Thank you for this comment, we agree, and have replaced the term expression with more accurate terminology throughout the manuscript.

4. Methods section. Table 1. While many of the internal standards are fine, in some cases they are not ideal e.g. for phosphatidylcholines. I think this is acceptable, as only relative concentrations are being reported, but some justification should be given.

We acknowledge the limitation of internal standards for absolute quantitation of certain species, which is indeed why the data are reported as relative concentrations in this study. The standards used were those we had available to us at the time.

5. I would have liked to see some representative chromatograms to get a better feel of the data. While quite a lot of lipids are reported, the number is not too overwhelming for representative chromatograms to be included in the supplementary.

We thank the reviewer for this suggestion. To increase the visual interpretability of the LC-MS methods, we have included schematic representations of the different compound classes' elution times throughout their respective chromatograms in Supplementary Methods Figure S5. Additionally, each lipid compound's individual retention time can be viewed in the Panorama repository https://panoramaweb.org/PD_Brain_Lipids.url.

6. It is very good to show all the MRMs in the supplementary, but perhaps some references could be included to help those not expert in shingolipid and phospholipid mass spectrometry to understand the transitions.

This is a good point from the reviewer and we have therefore included a statement clarifying that the MRMs represent characteristic fragmentation patterns for each species and also added the below reference regarding lipid analysis methods.

Han, X. *Lipidomics : comprehensive mass spectrometry of lipids*, (John Wiley & Sons, Incorporated, Hoboken, New Jersey, 2016).

Magny, R., et al. Lipid Annotation by Combination of UHPLC-HRMS (MS), Molecular Networking, and Retention Time Prediction: Application to a Lipidomic Study of In Vitro Models of Dry Eye Disease. *Metabolites* **10**(2020).

7. In Figure 1, it might help the reader if z-score was defined.

We agree that it would be beneficial to include the z-score's definition at its first mention and have thus added this to the legend of Figure 1.

8. Throughout the manuscript the term "n-hexosylceramide" is used, what does the "n" signify?

The authors apologise for not clarifying this. Apart from the mentioning on page 4 of n signifying the number of hexosyl groups, we have now also added this information to Table 1.

The statistics were beyond my expertise, but clearly there is a lot of valuable information here.

Reviewers' comments for the paper *Multi-Omic Analysis Reveals Lipid Dysregulation Associated with Mitochondrial Dysfunction in Parkinson's Disease Brain*

Comments from Reviewer #1

Major concerns

A major concern is still the very low number of brain samples analyzed. The authors have not added any samples in the revision process. There is only 1 sample for Braak stage 5-6 in Temporal cortex, parietal cortex, parahippocampus, cingulate cortex and caudate nucleus. The reviewer think that it is not possible to relate the data to stages of Parkinson's disease as done by the authors for example in figure 3. The authors only have data to compare controls with PD.

We appreciate the reviewer's concern regarding sample size. In response, we have acquired and analysed 40 additional PD samples to enhance the robustness of our dataset (please see Supplementary Table 1 for detailed information). Following normalisation and integration with the original dataset, this expanded analysis reinforces the original findings and increases the statistical power of our observations. Notably, several key changes have become more significant. Accordingly, we have revised Figures 2 and 3 to reflect the updated analysis and made corresponding modifications to the Results section. We have only performed analyses for the late-stage regions with five or more samples (cerebellum, frontal cortex and putamen in Figure 4).

In the "mid stage cases", where there are more brain samples studied, most lipid changes occur in cerebellum. Despite that the authors cite some articles on a role of cerebellum in Parkinson's disease, it is a region considered to have a minor influence of the clinical symptoms and signs of Parkinson's disease. It is difficult for the reviewer to see that these data bring any substantial new insight into the pathogenesis of Parkinson's disease.

We thank the reviewer for this thoughtful observation. In the newly expanded cohort, the previously observed lipid alterations in the cerebellum are now less pronounced, and as such, we have removed these observations from the revised manuscript to maintain focus on the most robust findings. Nonetheless, we believe the cerebellum remains of interest in the context of emerging literature, particularly with respect to lyso-phospholipid profiles. Our data suggest that the cerebellum is uniquely enriched in lyso-phospholipids, including lyso-phosphatidylcholine (LPC). Given recent studies proposing a potential neuroprotective role for LPC in modulating α -synuclein aggregation, we feel this observation merits further investigation. While not central to the current study, we believe this finding could offer a useful avenue for future research and have included it in the Discussion in this context.

Comments from Reviewer #2

This is a very interesting article showing how the phospholipid/sphingolipid profiles of human brain vary with location, age and disease state. The data will be of interest to clinicians, neuroscientists, biologists, biochemists and lipidologists. The variation of sphingolipid/phospholipid profiles in the different states of disease is fascinating, as is their clustering in different regions. The variation of lipid profile with Braak staging with further strengthens the manuscript. I only have minor comments and suggestions.

1. "Isoform" is not a word normally used in lipidomics I would recommend the authors use more standard nomenclature based on category (e.g. sphingolipids), class (e.g. ceramides, phosphosphingolipids), sub-class (e.g. sphingomyelins). See J. Lipid Res. 2005.46:839–861.

We thank the reviewer for highlighting the importance of precise nomenclature in lipidomics. In line with the reviewer's recommendation and the guidance provided in the referenced J. Lipid Res. 2005.46:839–861, we have reviewed the manuscript and replaced the term "isoform" with more appropriate and standard lipidomic terminology.

2. While it is excellent practice to deposit data in the Panorama Repository, it would be nice to see some real chromatograms in the Supplementary Results.

We thank the reviewer for this helpful suggestion. In response, we have added representative chromatograms to the Supplementary section to complement the deposited dataset. We believe this addition provides greater transparency regarding data quality and supports the robustness of our analytical approach. The chromatograms can be found in Supplementary Methods Figure S1.

3. The numbering of the Supplementary Figures seems to be out of sync with the main text.

We thank the reviewer for bringing this to our attention and apologise for the oversight. We have carefully reviewed and corrected the numbering of the Supplementary Figures to ensure consistency with the main text.

4. Methods. The authors could add to the text the solvent the brain tissue was homogenized in.

We thank the reviewer for this valuable suggestion. The solvent used for brain tissue homogenisation has now been clearly specified in the Methods section.

5. Methods. I suggest a few words could be added explaining the limitations of using non-category specific standards. As relative quantifications are used I don't think their use materially weakens the manuscript.

We appreciate the reviewer's insightful comment. We agree that the use of non-category specific standards introduces certain limitations, particularly with regard to absolute quantification and variability across lipid classes. We now acknowledge this limitation explicitly in the Methods section. However, as the reviewer notes, given that our analyses are based on relative quantification and consistent processing across all samples, we believe this approach remains valid for the purposes of comparative lipid profiling

6. What is the y-axis in Figure 4.

We thank the reviewer for their comment. In response, we have revised the figure and included clear, axes descriptions.

Besides these minor suggestions/comments, I see this as an excellent manuscript.